# Soft Wearable Robots: Development Status and Technical Challenges

**DOI:** 10.3390/s22197584

**Published:** 2022-10-06

**Authors:** Yongjun Shi, Wei Dong, Weiqi Lin, Yongzhuo Gao

**Affiliations:** State Key Laboratory of Robotics and System, Harbin Institute of Technology (HIT), Harbin 150001, China

**Keywords:** exosuit, wearable robot, soft exoskeleton, power assistance, human-robot interaction

## Abstract

In recent years, more and more research has begun to focus on the flexible and lightweight design of wearable robots. During this process, many novel concepts and achievements have been continuously made and shown to the public, while new problems have emerged at the same time, which need to be solved. In this paper, we give an overview of the development status of soft wearable robots for human movement assistance. On the basis of a clear definition, we perform a system classification according to the target assisted joint and attempt to describe the overall prototype design level in related fields. Additionally, it is necessary to sort out the latest research progress of key technologies such as structure, actuation, control and evaluation, thereby analyzing the design ideas and basic characteristics of them. Finally, we discuss the possible application fields, and propose the main challenges of this valuable research direction.

## 1. Introduction

As many task scenarios have high requirements on physical capability, human beings desire to break through physiological limits by utilizing an external mechanical device so that they can realize goals such as less fatigue, fast movement or great strength [1]. For groups with impaired motor ability, it is also hoped that a similar system can achieve the effect of rehabilitation training [2]. Driven by the practical demands above, research on related technologies of wearable robots have begun to flourish, which facilitates the formation of relatively systematic theoretical knowledge. These robots belong to electromechanical integration systems closely cooperating with the wearers. They analyze the human intention through multi-source sensors, solve the motion commands with control strategies designed according to specific requirements, adjust different forms of actuators to generate suitable power, and finally complete actions in collaboration with human joints.

In this article, the connotation of wearable robot is consistent with that of *exoskeleton* defined in ASTM F48 [3], which “*augments, enables, assists, and/or enhances physical activity through mechanical interaction with the body*”, and “*may include rigid or soft components, or both*”. Rigid systems take advantage of link mechanisms parallel to the limbs to achieve load transmission and joint assistance [4]. Their basic characteristics bring about many intractable problems. First of all, the overall volume often seems pretty large, causing inconvenience of operation and carrying. Additionally, human joints possess a relatively complex anatomical structure, whose rotation center is constantly changing within the range of motion. This phenomenon makes it difficult for exoskeletons that simulate human joints with rotating pairs to reproduce the natural trajectory of limb movement. The misalignment between the joints of an exoskeleton and wearer seriously weakens the man–machine coordination. Furthermore, the exoskeleton tends to possess large mass and inertia, due to the extensive usage of rigid structures. On the one hand, it inevitably adds impedance to the whole system, while on the other hand, it also introduces additional load that needs to be borne, resulting in reduced wearing comfort and increased metabolic cost.

Under the restrictions of the disadvantages above, many researchers are turning to the field of soft wearable systems, trying to achieve a lightweight and flexible scheme with the help of a soft medium for power transmission. The soft systems, usually described as exosuits or soft exoskeletons, do not rely on the linkage mechanisms to assist joint movements, and have no rigid frame to support the load [5]. Simulating the working principle of muscles and tendons, they apply flexible driving units installed on the body surface to transmit power to corresponding joints. The exosuits avoid many defects of rigid exoskeletons from the design principle, and show greater performance in terms of comfort, convenience and coordination [6].

In recent years, many cutting-edge technologies and various new prototypes have emerged with research on exosuits taking an upward trend [7,8]. Numerous review literature collates and compares the technological achievements in this field from different emphases. For example, some scholars focus on soft wearable robots for lower limbs based on a cable-driven scheme and divide them into three categories for discussion [9]. Some papers systematically describe soft and rigid systems that assist certain local joints and comprehensively analyze the technical characteristics of both [10]. There are also some studies that introduce exosuits according to the target assisted joints [11].

In this paper, we aim to comprehensively review the current development status of soft wearable robots for various human joints, summarize the key technologies supporting the entire system, and discuss the existing challenges that hinder their further development. In Section 2, according to the different parts that need assistance, all soft wearable systems could be divided into three categories, namely upper extremity exosuits targeting shoulder, elbow and wrist, lower extremity exosuits aiming at the hip, knee and ankle, and back-assist exosuits for lumbar support. In Section 3, from the four aspects including structure, drive, control and evaluation, we review the design concepts and technical solutions which have been put forward by relevant scholars to realize power assistance with soft wearable robots. In Section 4, this paper discusses the main application scenarios of this technology in the future and analyzes the problems that need to be solved.

In general, this review will report the current progress and typical results and provide a basic reference and a systematic introduction to those who want to investigate exosuit research.

## 2. Overview of Exosuit

There are significant differences in the anatomical structures and movement patterns for each joint. Different assistance schemes should be carefully evaluated according to specific application requirements. This section will be described using aspects of upper extremity exosuits, lower extremity exosuits and back-assist exosuits.

### 2.1. Upper Extremity Exosuits

The upper extremity system simulates the working principle of muscles and tendons to generate power at the shoulder, elbow or wrist, which assists the upper arm, forearm or hand to perform specific movements or pick up loads. Several representative exosuits are shown in Figure 1.

#### 2.1.1. Shoulder

The shoulder connects the trunk and upper arm, which consists of the humerus, clavicle and scapula to form its basic skeleton. It belongs to a compound joint, including scapulothoracic, acromioclavicular, glenohumeral and sternoclavicular. This complex structure enables the shoulder to have excellent movement flexibility in space and can support a variety of action modes such as flexion, extension, adduction, abduction, internal rotation, external rotation and circumduction. Table 1 lists key features of some exosuits for shoulder assistance.

Much literature focused on the assistance of one action mode of the shoulder to verify the effect and feasibility of their schemes. To help people with cerebral palsy perform repetitive task practice, Natividad et al. [12] designed a lightweight and unrestrictive exosuit for shoulder abduction. An inflatable beam acted as the driving unit, whose position control was realized by regulating internal pressure. Thompson et al. [13] described the basic design philosophy of a cable-driven exosuit with soft pneumatic actuators for augmenting shoulder flexion. They took advantage of fiber-reinforced elastomeric enclosures to create two actuation architectures including the nested linear form and pennate form, and then compared their output characteristics. The driving force was transmitted to the corresponding joint via a Bowden cable. O’Neill et al. [14] developed an abduction-assisted system that placed a textile-based pneumatic actuator under the armpit. The main purpose was to study how structural parameters of the actuator affected its moment characteristics.

**Figure 1 sensors-22-07584-f001:**
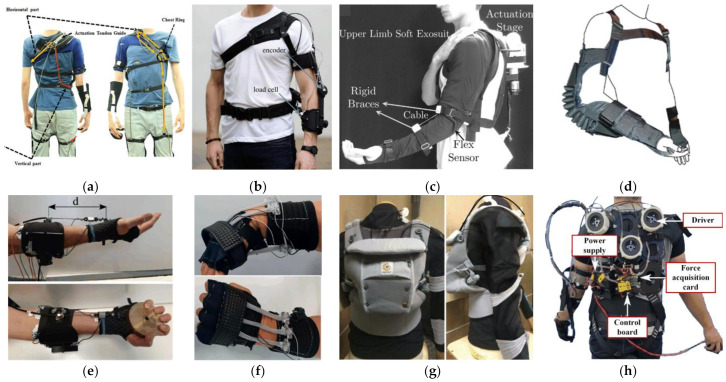
Exosuits for upper limbs: (**a**) Unpowered shoulder exosuit for supporting weight and relieving muscle fatigue [15]; (**b**,**c**) Wearable devices based on Bowden cables for elbow assistance [16,17]; (**d**) Pneumatic exosuit for load holding and carrying [18]; (**e**) Cable-driven device for wrist flexion [19]; (**f**) Wearable robot using shape memory alloy [20]; (**g**) An exosuit with a combination of different textiles through force-compliant sewing [21]; (**h**) A cable-driven prototype to assist shoulder flexion/extension, shoulder adduction/abduction, and elbow flexion/extension [22].

Some research achieved simultaneous assistance for multiplanar movements of the shoulder through relatively sophisticated design. O’Neill et al. [23] utilized two types of soft pneumatic actuators to provide power for abduction and horizontal flexion/extension. The main parts were nearly invisible when folded and not in use and can also be put on and taken off as easily as ordinary clothes. By comparing biological muscle activity before and after powering on the system, it was proved that this wearable robot can make a positive contribution to shoulder movement in daily life. Park et al. [15] proposed an unpowered wearable device for flexion/extension and adduction/abduction, which can provide gravity compensation in any posture. The passive actuator applied assistive torques to the joint through flexible tendons. Evaluation experiments based on electromyography (EMG) measurement proved that muscle fatigue can be effectively relieved when performing repetitive limb movements. Varghese et al. [24] put forward a novel perception strategy for a shoulder exosuit with multiple degrees of freedom (DoFs). The research aimed at establishing a tendon-routing architecture inspired by muscle synergies and using multivariate multiple regression to estimate joint angle based on sensor data.

**Table 1 sensors-22-07584-t001:** Several representative exosuits for shoulder assistance.

Exosuit/Study	Year	Movements	Actuation	Sensors	Control	Function
Natividad et al. [12]	2016	Abduction	Pneumatic actuator	Three-axis accelerometer, pressure sensor	Position control	Healthcare
Park et al. [15]	2017	Flexion/extension, adduction/abduction	Passive actuator	-	-	Physical enhancement
O’Neill et al. [23]	2017	Abduction, horizontal flexion/extension	Pneumatic actuator	6-axis load cell, encoder, pressure sensor	Open loop control	Healthcare
Thompson et al. [13]	2019	Flexion	Pneumatic actuator with Bowden cable	Load cell, pressure gauge	Proportional-integral-differential (PID) controller	Physical enhancement
Varghese et al. [24]	2020	Multi-DoF movements	Cable-driven module	Tendon-based sensing units	Feedback control	Healthcare
O’Neill et al. [14]	2021	Abduction	Pneumatic actuator	External pressure sensor, torque sensor	Pressure control	Physical enhancement

#### 2.1.2. Elbow

The basic configuration of the elbow contains the ulna and radius in the forearm and the humerus in the upper arm. The biceps, triceps, brachioradialis and other muscle groups work together to control the corresponding joint movement. It is often regarded as a rotational pair, with one DoF in flexion/extension. Table 2 introduces some research related to elbow exosuits.

In terms of elbow assistance, Masia’s team conducted relatively comprehensive studies on soft wearable robots driven by Bowden cables [16,25]. In [26], an easily scalable embedded architecture was mentioned, which could be used for power management, low-level motor control, high-level signal processing and data streaming. To make the cable-driven module compact and improve its transmission efficiency, a scheme of driving two Bowden cables through a single motor was adopted, which can provide additional torque for joint flexion and extension in an agonist–antagonist manner [27]. Later, they even invented a more complex actuation mechanism, trying to drive multiple DoFs through only one prime mover, so that it can realize the assistance for elbow joints of both arms [28]. The exosuit equipped with this module utilized a novel PID-modulated pulse width modulation (PWM) controller to precisely regulate the velocity for each DoF independently [29].

Pneumatic wearable robots also looked suitable for providing elbow assistance. Thalman et al. [30] used an array of air chambers encased in nylon fabric to build an exosuit for repetitive lifting tasks. A mathematical model for bending behavior and torque calculation was developed. Nassour et al. [18] adopted a similar structural principle to design a powered elbow exosuit. Their experiments on 12 individuals demonstrated that this exosuit can significantly reduce muscle activity, metabolic consumption and fatigue, which meant that it was appropriate for many tasks including lifting, holding and carrying. In addition, Ang et al. [31] proposed a wearable system for assisting elbow flexion based on a bellow-type soft pneumatic actuator. A model was built to estimate the maximum tip force of this 3D printed actuator, which could provide a reference for adjusting its parameters to match different payloads. Through shape deposition manufacturing methods, Gao et al. [32] built a soft pneumatic elbow exosuit with silicone rubber, and implanted three sensors (barometric pressure sensor, bending sensor and force sensor) into the system. The relationship between barometric pressure and bending value was obtained through several experiments, and it was also shown that the root mean square of the EMG signal can be reduced by using this system.

In contrast, there was very little research on how to apply twisted string actuators to an elbow exosuit. Hosseini et al. [33] made attempts to demonstrate the possibility of this application. With two twisted string actuators generating power independently, the proposed wearable device can assist movements for single- or dual-arm mode under load conditions. A controller based on surface electromyograph (sEMG) took charge of activating and regulating the system output to save human efforts.

**Table 2 sensors-22-07584-t002:** Several representative exosuits for elbow assistance.

Exosuit/Study	Year	Movements	Actuation	Sensors	Control	Function
Dinh et al. [34]	2017	Flexion/extension	Cable-driven module	Flex sensor, two load cells	Hierarchical control	Physical enhancement
Gao et al. [32]	2017	Flexion	Pneumatic actuator	Pressure, bending and force sensors	-	Healthcare
Chiaradia et al. [26]	2018	Flexion/extension	Cable-driven module	Stretch sensor, load cell	Gravity compensation control	Physical enhancement
Thalman et al. [30]	2018	Flexion	Pneumatic actuator	Pressure sensor	Open-loop control	Physical enhancement
Lotti et al. [35]	2020	Flexion	Cable-driven module	Load cell, encoder, EMG electrode	Model-based myoelectric control	Physical enhancement
Ang et al. [31]	2020	Flexion	Pneumatic actuator	-	-	Physical enhancement
Hosseini et al. [33]	2020	Flexion	Twisted string actuator	Force sensor, encoder, sEMG sensor	sEMG-based control	Physical enhancement
Nassour et al. [18]	2021	Flexion	Pneumatic actuator	Pressure sensor	Switch operated by a person	Physical enhancement

#### 2.1.3. Wrist

The wrist connects the human hand to the forearm, and belongs to the compound joint composed of the articulationes carpometacarpeae, articulatio mediocarpalis, articulus radiocarpicus and articulationes radioulnaris distalis. It can complete many actions such as flexion, extension, abduction and adduction, which are assisted by muscles including the flexor carpi radialis, extensor carpi ulnaris, extensor carpi radialis, and flexor carpi ulnar. Table 3 details a few studies on wrist exosuits.

Some research has focused on how to construct a wrist exosuit with different DoFs by combining many pneumatic artificial muscles. The prototypes in [36,37] both adopted an arrangement with two actuators located at the upper and lower parts of a sleeve. The former applied air bags to keep the wrist in a neutral position for a typing application, while the latter used fold-based actuation mechanism to complete motions in two DoFs. A wearable robot named EXOWRIST [38,39] tried to help perform wrist rehabilitation training with four pneumatic actuators symmetrically arranged around the forearm. It achieved independent movement assistance for extension, flexion, ulnar deviation and radial deviation by inflating two designated actuators and deflating the remaining ones. A PID-based algorithm was responsible for motion control in performance evaluation experiments. Bartlett et al. [40] proposed a design scheme that arranged two actuators crossed in pairs on the upper and lower sides of wrist. With an ordered combination of their outputs, it was possible to support almost all action modes, especially supination and pronation. Additionally, Al-Fahaam et al. [41] alternated three contraction muscles and two extensor bending muscles above the wrist to assist joints in all DoFs.

Cable-driven modules also performed very well for this type of system. Exo-Wrist [42] was used to facilitate patients’ participation in constraint-induced movement therapy, and strengthened the paretic wrist to complete a dart-throwing motion. This cable-driven device equipped with a corset active anchor would compress the forearm only when necessary, consequently improving safety and comfort. Chiaradia et al. [19] launched a soft exosuit for wrist flexion, which contained a glove that could increase transmission efficiency and improve pressure distribution.

**Table 3 sensors-22-07584-t003:** Several representative exosuits for wrist assistance.

Exosuit/Study	Year	Movements	Actuation	Sensors	Control	Function
EXOWRIST [38]	2015	Extension/flexion, ulnar/radial deviation	Pneumatic actuator	Linear flex sensor, pressure sensor	PID-based control	Healthcare
Bartlett et al. [40]	2015	All DoFs	Pneumatic actuator	Pressure sensor	-	Healthcare
Al-Fahaam et al. [41]	2016	All DoFs	Pneumatic actuator	Pressure sensor	Direct pressure control	Healthcare
Zhu et al. [36]	2017	Flexion/extension	Pneumatic actuator	Inertial measurement unit (IMU), pressure sensor	-	Healthcare
SWS [37]	2019	Flexion/extension, radial/ulnar deviation	Pneumatic actuator	Pressure sensor	-	Healthcare
Exo-Wrist [42]	2019	Dart-throwing motion	Cable-driven module	Load cell	-	Healthcare
Jeong et al. [20]	2019	Extension/flexion, ulnar/radial deviation	Shape memory alloy-based actuator	-	-	Healthcare
Chiaradia et al. [19]	2020	Flexion	Cable-driven module	IMU, force sensor	Admittance controller	Physical enhancement

There were also some scholars who attempted to study shape memory alloy-based wearable robots for the wrist. Jeong et al. [20] made use of this material to fabricate highly stretchable actuators, and applied five of them to achieve assistance for multiple actions including flexion, extension, ulnar deviation and radial deviation. To improve the system response speed, they further investigated the active cooling system with coolant circulation, and explored the application effects of different coolants [43].

#### 2.1.4. Multi-Joints in the Upper Extremity

When the human arm is executing typical daily movements, it often requires close coordination of the shoulder, elbow and wrist to generate power. Therefore, selecting certain joints for assistance and designing a more comprehensive upper extremity system should be of great significance for improving versatility and efficiency. Some works on multi-joint exosuits are shown in Table 4.

CRUX [44] combined seven cables for power transmission and six micromotors for torque generation to achieve strength amplification during humeral rotation, elbow flexion/extension, wrist pronation/supination, lateral shoulder raise, and forward shoulder raise/lower. It determined the optimal cable paths of minimal extension by conducting measurement experiments and could be operated using either a two-axis analog joystick or a closed-loop controller. Paired with virtual reality technology, it was applied to a novel rehabilitative experience called Project Butterfly [45]. Relevant experiments demonstrated its performance including feasibility, ease of use and comfort. Samper-Escudero et al. [21] and Shi et al. [22] also developed cable-driven wearable systems to provide assistance for both shoulder and elbow.

After developing a flexible driving structure named “18 Weave” with thin McKibben muscle, Abe et al. [46] constructed an exosuit based on it to support upper limb movements such as shoulder flexion and elbow flexion. The PowerGrasp [47] belonged to a modular pneumatic system which adjusted the chamber’s stiffness to support shoulder, elbow, and wrist/finger during assembly. Authors have carried out research on automated fatigue detection and classification to give advice on rest or to adjust support for workers.

Auxilio [48] was a lightweight exosuit equipped with three twisted string actuators, which offered dynamic rehabilitation of shoulder flexion, shoulder abduction, and elbow flexion. The relevant authors detailed the working principle and carried out analyses and experiments on kinematics and force.

**Table 4 sensors-22-07584-t004:** Upper extremity exosuits for multi-joint assistance.

Exosuit/Study	Year	Movements	Actuation	Sensors	Control	Function
CRUX [44]	2017	Assistance for shoulder, elbow and wrist	Cable-driven module	IMU	Human-in-the-loop control or closed-loop	Healthcare
Auxilio [48]	2017	Shoulder flexion/abduction, elbow flexion	Twisted string actuator	Motion sensor device	Mirror therapy	Healthcare
Abe et al. [46]	2019	Shoulder flexion and elbow flexion	Pneumatic actuator	Pressure sensor	-	Physical enhancement
Samper-Escudero et al. [21]	2020	Flexion of shoulder and elbow	Cable-driven module	Flexion sensor, encoder	Sliding mode controller	Physical enhancement
PowerGrasp [47]	2021	Assistance for shoulder, elbow and wrist	Pneumatic actuator	IMU	Adaptive pose-dependent control	Physical enhancement
Shi et al. [22]	2022	Flexion/extension for shoulder and elbow, adduction/abduction for shoulder	Cable-driven module	IMU, encoder, tension sensor	Torque estimation-based Control	Healthcare

### 2.2. Lower Extremity Exosuit

The lower extremity system needs to provide assistance for the hip, knee or ankle to help a user walk or climb stairs. Several representative exosuits are shown in Figure 2.

#### 2.2.1. Hip

The human hip belongs to a typical enarthrosis that connects the lower extremity to the torso. It consists of the femoral head and acetabulum in physiological structure and can perform actions along multiple axes including flexion/extension, adduction/abduction, and internal/external rotation. There are several representative hip exosuits in Table 5.

The hip exosuit proposed by Walsh and his team preferred to assist joint extension at first. In [49], a portable wearable robot was designed with a spooled-webbing actuator mounted onto the back. Ribbons attached to thigh braces could produce about 30% of natural joint torque during level-ground walking. In [50], they described implementation details of a mono-articular untethered system with Bowden cables and mobile actuators. In order to explore its effect on metabolic cost and retention of metabolic improvement, some subjects participated in multiple training sessions to demonstrate energetic adaptation. Much research has been carried out by using tethered cable-driven systems which usually needed to work together with the treadmill and the off-board actuation platform [57,58,59,60]. Although this implementation sacrificed portability, it could reduce the additional load borne by a person, thereby improving assistive efficiency and wearing comfort. Afterwards, in 2022 they designed a lightweight exosuit for hip flexion and applied human-in-the-loop optimization to individualize assistance, achieving significant reduction in metabolic expenditure [51].

**Table 5 sensors-22-07584-t005:** Several representative exosuits for hip assistance.

Exosuit/Study	Year	Movements	Actuation	Sensors	Control	Function
Asbeck et al. [49]	2015	Extension	Spooled-webbing actuator	Load cell, encoder, footswitches	Position control	Military application
Jin et al. [61]	2017	Flexion	Spooled-webbing actuator	Load cell, gyroscope	Tension force control	Physical enhancement
John et al. [62]	2017	Multiple DoFs	Cable-driven module	Force sensor	Proportional feedback velocity controller	Physical enhancement
Haufe et al. [63]	2020	Flexion	Passive actuator	-	-	Physical enhancement
Yang et al. [64]	2021	Flexion	Passive actuator	Load cell	-	Physical enhancement
Chen et al. [65]	2021	Flexion	Cable-driven module	IMU, load cell	Gait identification, admittance controller, position controller	Physical enhancement
Kim et al. [51]	2022	Flexion	Cable-driven module	IMU, load cell	Force control based on Human-in-the-loop optimization	Physical enhancement
Yang et al. [66]	2022	Abduction	Cable-driven module	IMU, load cell	High and low-level controller	Healthcare
Tricomi et al. [67]	2022	Flexion	Cable-driven module	IMU	Adaptive oscillators-based control	Physical enhancement

In terms of hip flexion assistance, there were many different options for providing desired joint torque. Jin et al. [61] proposed a soft wearable robot that utilized winding belts for power transmission, and then conducted evaluation experiments of metabolic cost on occasions when elderly persons were implementing long-distance level and inclined walking [68]. Some researchers made full use of elastic elements and textile materials to build passive exosuits and explored their biomechanical effects during walking [63] or running [64]. The prototype based on Bowden cables could deliver the motor output along a fixed path to the anchor point of the anterior thigh, and lead to a metabolic reduction effect as well [65,67].

In [66], an exosuit for augmenting hip abduction was proposed to maintain the coronal angle of the pelvis basically stable. It aimed at counteracting adverse impacts of external knee adduction moment.

Obviously, it was quite necessary to simultaneously assist the multiple actions of the hip. John et al. [62] presented a wearable prototype which incorporated a novel concept of a cross-wire structure. Through selectively driving a subset of four Bowden cables arranged in a cross form, it generated joint torque in six directions to strengthen multi-DoF hip movements. Subsequent studies demonstrated that this system could be applied to induce turning and control walking direction for the wearer [69].

#### 2.2.2. Knee

As the largest joint and one of the most complex ones in the human body, the knee consists of the lower femur, upper tibia and patella, and is reinforced by ligaments. Its motor function is mainly reflected in flexion and extension. Considering that damage may occur easily under large loads, it makes sense to provide assistance for knee movements. Table 6 describes some exosuits that assist the knee joint.

Applicable to help the elderly go upstairs in ordinary living, a soft suit called Hitexosuit [70] played a crucial part in enhancing knee extension. Its twisted string actuator simulated muscle contraction to generate force, which could assist in propelling the body upwards at just the right moment.

Park et al. [71] developed a cable-driven exosuit that integrated flexible textiles and a rigid frame. The main purpose of using rigid components was to extend the momentarm for knee extension. To support knee extension and flexion for stair ascent and descent, Lee et al. [72] proposed a system with a series elastic actuator activating both front and rear wires. On the one hand, this design scheme enhanced the flexibility and compliance of wire during power transmission, while on the other hand, it could obtain the interaction force between human and robot by detecting spring deformation.

With respect to pneumatic knee exosuits, Sridar et al., carried out a series of exploratory studies. In [76], the authors fabricated two soft-inflatable actuators with different cross-sectional shapes to compare their output performance, and picked out the I-shape to build a wearable robot for knee extension rehabilitation. Afterwards, they detailed how to detect human gait and control actuator stiffness, consequently generating the desired torque during the swing phase [77]. In [73], an untethered exosuit was proposed which adopted inflatable actuator composites instead of inflatable fabric beams. Benefiting from its higher actuation speed and lower energy loss, a portable pneumatic source could be applied to the wearable system. In addition, referring to the accordion structure, Fang et al. [74] designed a system based on foldable pneumatic bending actuators. A wearable device named SLAK [75] applied pleated pneumatic interference actuators to produce high torque for sit-to-stand tasks.

#### 2.2.3. Ankle

The ankle contributes significantly to some physiological functions, such as weight support, stability maintenance and impact bearing. Its main DoFs include plantarflexion/dorsiflexion, varus/valgus, and internal/external rotation. As shown in Table 7, many scholars have adopted a variety of methods to strengthen this human joint.

The team, led by C. J. Walsh, also conducted continuous research on cable-driven unilateral exosuits for ankle rehabilitation which could augment plantarflexion and dorsiflexion [78,79]. According to their description, this system had a positive effect on improving forward propulsion and ground clearance so that it could help stroke patients perform more natural and economical locomotion [80]. They not only described the biological mechanisms about exosuit-induced metabolic reductions, which seemed related to the center of mass (COM) power produced by each limb [81], but also pointed out that physical interface dynamics had great influence on power transmission and human augmentation benefits [82]. To accommodate variability during hemiparetic walking, specialized schemes such as offline assistance optimization [83] and muscle-based assistance [84] took charge of customizing the assistance profile. Some evaluation experiments [85,86] were carried out with stroke survivors, demonstrating that this system could lead to an increase in walking speed/distance and a decrease in metabolic expenditure. The exosuit designed by Schubert et al. [54] transmitted power to the ankle through utilizing Bowden cables, and realized plantarflexion support and dorsiflexion augmentation based on autonomous gait recognition.

**Table 7 sensors-22-07584-t007:** Several representative exosuits for ankle assistance.

Exosuit/Study	Year	Movements	Actuation	Sensors	Control	Function
Park et al. [87]	2011	Dorsiflexion, inversion and eversion	Pneumatic actuator	Strain sensor, IMU, pressure sensor	Feed-forward controller or feedback proportional controller	Healthcare
Bae et al. [79]	2018	Plantarflexion and dorsiflexion	Cable-driven module	IMU, load cell	Hierarchical closed-loop controller	Healthcare
ExoBoot [88]	2018	Plantarflexion	Pneumatic actuator	Pressure sensors, IMU	Open loop pressure controller	Physical enhancement
Thalman et al. [89]	2019	Lateral/medial support, dorsiflexion,	Pneumatic actuator	Force sensitive resistor, fluidic pressure sensor	Bang-bang control	Healthcare
Yandell et al. [90]	2019	Plantarflexion	Passive actuator	-	-	Physical enhancement
Siviy et al. [83]	2020	Plantarflexion	Cable-driven module	IMU, encoder, load cell	Admittance controller	Healthcare
Nuckols et al. [84]	2021	Plantarflexion	Cable-driven module	IMU, load cell	PI force control loop cascaded with current loop	Physical enhancement
Schubert et al. [54]	2021	Plantarflexion and dorsiflexion	Pneumatic actuator with Bowden cable	Force sensing resistor, IMU	Bang-bang control	Healthcare

Inspired by muscle-tendon architecture, an active soft orthotic device proposed in [87] applied three pneumatic artificial muscles to help perform ankle dorsiflexion, inversion and eversion. The ExoBoot [88], a soft inflatable robot, made use of a textile-based actuator installed on foot and shin, and produced assistive torque for plantarflexion. Thalman et al. [89] developed a sock-like pneumatic exosuit to help impaired individuals restore natural gait. Its front-mounted actuator aimed at enhancing dorsiflexion within the swing phase, while the variable stiffness actuators fixed on the lateral and medial sides could offer support during heel strike.

Yandell et al. [90] proposed an unpowered prototype configured seamlessly underneath clothing, which blended perfectly with the human body. Arranged parallel to the gastrocnemius, a stiff assistance spring acted to assist plantarflexion from early stance to late stance and was released during the leg swing to avoid impeding joint motion. An under-the-foot clutch mechanism realized the automatic switch between its engagement and disengagement according to periodic friction generated by plantar pressure.

#### 2.2.4. Multi-Joints in the Lower Extremity

Obviously, the coordinated output of hip, knee and ankle ensures the effective execution of lower limb movements. Although assisting multiple joints seems more in line with the application requirements, it may make the system rather complex and heavy. Therefore, relevant research has always paid attention to the balance between lightweight and high performance. Some lower extremity exosuits that provide multi-joint assistance are presented in Table 8.

Research of Walsh and his team aimed to study cable-driven exosuits for both hip and ankle. In [95,96], a portable prototype combined the suit composed of multiple webbings with Bowden cables extending from the pelvis to the heel, which achieved the synthesis of passive and active assistance. Apart from this multiarticular load path for ankle plantarflexion and hip flexion, a separate load path for hip extension was added to an improved wearable system [97]. With the support of this design, they verified the individual parameter tuning method based on positive ankle augmentation power [98] and conducted evaluation experiments under conditions where the device was powered, unpowered or unpowered with equivalent mass removed [1]. Another commonly used system possessed a combined form that enhanced ankle plantarflexion directly with an off-board cable-driven platform and supported hip flexion indirectly through textile architecture. Based on this, they utilized a method to independently control its assistance level during the positive-power phase and the negative-power phase [99] and compared the performance between an ankle moment inspired technique and an ankle positive power inspired technique [100]. The relationship between assistance magnitude and metabolic cost was also established, while simultaneously exploring the variation of underlying gait mechanics [92]. In addition to the studies above, they also proposed a pneumatic exosuit based on McKibben actuators, a triangulated webbing network and an off-board compressor, which aimed to augment hip, knee and ankle simultaneously [91].

Myosuit [93] adopted a biarticular architecture to work against gravity, with active tendons and passive ligaments, respectively, in charge of hip/knee extension support and hip/knee flexion assistance. The load-intensive activities of daily living including sitting transfers could benefit from a closed-loop force controller for gravity compensation. Relevant researchers confirmed the safety and feasibility of activity-based training with this wearable device [101], and studied the process that new users adapted for robotic assistance [53].

The European Consortium planned to develop a modular lower extremity exosuit (XoSoft) targeting people with partial mobility limitations. Based on a first prototype which assisted the flexion of hip and knee, the basic functionality was analyzed from aspects of practicability, usability, comfort and kinematics [102]. Subsequently, Beta 1 prototype [56] introduced a quasi-passive actuation concept to transfer energy between different gait phases with the combination of electromagnetic clutch and elastic band. Later, an upgraded version known as the Gamma prototype [94] adopted pneumatic quasi-passive actuation which contained a textile-based clutch actuated by vacuum pressure, and provided relatively comprehensive assistance for hip flexion/extension, knee flexion/extension and ankle plantarflexion/dorsiflexion.

### 2.3. Back-Assist Exosuit

The back-assist exosuit attempts to relieve biomechanical loads on the lumbar spine through active or passive assistance. It has the capability to help perform physically demanding tasks and prevent the wearer from suffering back pain or injury. Several representative prototypes are described in Table 9.

In terms of back assistance, many studies preferred to develop exosuits with passive actuators, which present advantages such as simplicity, practicality and lightness. A biomechanically assistive garment proposed in [103] distributed the load from the waist to the shoulder and thigh, with two crossed elastic bands connecting the upper- and lower-body interfaces. Exosuit proposed in [104] (see Figure 3a), HeroWear Apex [105] (see Figure 3b) and Auxivo LiftSuit [106] followed a similar design concept, and then introduced the mode switching method that provided lumbar assistance when engaged and maintained full slack when disengaged. Additionally, the BASE emulator [107] was developed to analyze massive design factors of back-assist exosuits, equipped with onboard load cells for interaction force detection, sEMG electrodes to obtain muscle activities, and a modular moment arm that allowed testing for different configurations of passive actuation.

Active devices could generate arbitrary assistance as needed and did not require the process of energy storage. The AB-Wear suits developed by Inose et al., could be divided into endoskeleton types [108] and semi-endoskeleton types [109]. The former contained two artificial muscles, a balloon actuator and an amplification mechanism, while the latter was additionally equipped with reduction mechanism to reduce compressive forces borne by the human back. Govin et al. [110] designed a soft robotic orthosis to help patients maintain good posture and stabilize the lumbosacral spine. Two pneumatic bladders arranged in parallel make it possible to exert enough assistive torque on the back. Furthermore, the prototype in [111] relied on the combination of a hyper-redundant continuum mechanism and tethered cable-driven platform to support the waist without restricting natural movements. ABX [112] applied two Bowden cables with an “X” pattern to provide torque for multiple DoFs of the lumbosacral joint. Its control system containing a finite state machine and force controller was responsible for state pattern detection, cable tension update, motor velocity calculation and so on. In addition, the soft power suit equipped with twisted string actuators could also be well qualified for assisting trunk movements [113].

**Table 9 sensors-22-07584-t009:** Several representative exosuits for back assistance.

Exosuit/Study	Year	Movements	Actuation	Sensors	Control	Function
AB-Wear [109]	2017	Trunk extension	Pneumatic actuator	Load cell, pressure sensor	External operator	Physical enhancement
Biomechanically assistive garment [103]	2018	Trunk extension	Passive actuator	-	-	Physical enhancement
Govin et al. [110]	2018	Trunk extension	Pneumatic actuator	IMU	Control based on position of spine	Healthcare
Yang et al. [111]	2019	Trunk extension	Cable-driven module	Load cell, IMU	Virtual impedance control strategy	Physical enhancement
Yao et al. [113]	2019	Trunk extension	Twisted string actuator	Force sensor, IMU	Control according to motion intention and conditions	Physical enhancement
Lamers et al. [104], HeroWear Apex [105]	2021	Trunk extension	Passive actuator	-	-	Physical enhancement
Auxivo LiftSuit [106]	2022	Trunk extension	Passive actuator	-	-	Physical enhancement
BASE emulator [107]	2022	Trunk extension	Passive actuator, moment arm	Load cell, surface EMG electrodes	-	Physical enhancement
ABX [112]	2022	Flexion/extension, axial rotation, and lateral bending	Cable-driven module	Load cell, IMU	Two-tier control containing finite state machine and force controller	Physical enhancement

## 3. Key Technologies

The soft wearable robot is a type of mechatronic equipment which integrates multiple subsystems and research points. This section will focus on the principles and implementation of the relevant key technologies.

### 3.1. Material and Structure

Relevant studies seldom reveal the underlying design principles of wearable textile architecture. If not optimized accordingly, it will seriously weaken the interaction comfort and transmission efficiency.

For powered exosuits, it is necessary to maximize the fabric stiffness along the direction of force transmission [96]. The force on the flexible medium is delivered to human joints through bindings. Bindings with high elasticity will deform greatly during this process, consequently reducing the effective displacement of flexible medium and absorbing part of the energy provided. Therefore, it will make the control error increase, and eventually significantly weaken effectiveness of power assistance. To prevent this, high-stiffness fabrics should be used to customize the stressed parts of the wearable suit and should fit tightly to the body to avoid wrinkles. For example, CRUX [44] adopted a neoprene base layer covering the torso and arms, which could maintain enough rigidity for anchoring on-board components and actuation forces.

Efforts should be made to guarantee comfort while wearing [114]. Excessive normal pressure and shear force between the wearer and suit usually make people feel uncomfortable. Exosuits with such undesirable properties will accelerate the human body into a state of fatigue and even lead to poor blood circulation. Localized shear force at the skin surface can generally be minimized by optimizing force transmission, while normal pressure will be reduced by increasing the contact area.

It is recommended to appropriately utilize the physiological structure of the human body to fix the end of the textile. If the friction between the binding and body surface becomes insufficient during the man–machine interaction, the former may move up and down along the limb or torso under the drive of the external force. Continuous reciprocating sliding will possibly lead to skin damage and is not conducive to improving system performance. Using the specific body structure to limit the bindings seems to be an effective solution. For example, for a lower extremity system that transmits power along the entire leg, we can choose the pelvis and feet as fixed positions and connect several straps and bindings in series between them; for upper extremity systems, textiles can be attached between the vest on the torso and the glove on the hand to ensure the relative stability of the position between the suit and wearer.

### 3.2. Actuator and Power Transmission

According to the review of research status, the commonly used driving units of soft wearable robot can be divided into the following categories. This section aims at describing the working principles and features, and analyzing their advantages and disadvantages in applications.

**Cable-driven module:** Such a device often utilizes a combination of cable and sheaths to transmit power from the prime mover to corresponding joints [95]. The sheath installed on the body surface can be set directly according to any desired path between the actuator and anchor point, thereby constraining retraction and release of inner Bowden cables along the fixed route. This mechanism facilitates the flexible configuration of the structural layout and enhances the compactness of the human–machine integration [115]. When no assistance is provided, the slack Bowden cable will not substantially impede the wearers’ movement, as if they put on a normal piece of clothing.

However, there are some inherent problems with this wearable application of Bowden cables. The friction generated by the relative movement between cable and sheath will hinder the power transmission process. It leads to dissipation of part of the mechanical work and reduction in the assistive efficiency. In addition, the cable slack will delay the force transmission to joints, which exacerbates the response hysteresis and nonlinear behavior of the system [17].

**Pneumatic actuator:** When the internal air pressure of this device is adjusted, the volume or shape changes accordingly, thereby generating torques on the human joint connected with it. Generally, it’s necessary to apply an air source to complete the inflation or deflation [73]. During this process, the pneumatic actuator will execute actions such as elongation, shortening or bending. If it is properly coordinated with the limb movements, assistance can be provided at the corresponding human joint.

A pneumatic actuator has the advantage of a high power-to-weight ratio, so that it can adapt to application scenarios with a large joint load. For example, SLAK could generate a maximum torque of 324 Nm at the knee, exceeding the peak value required for sit-to-stand motion [75]. However, most pneumatic exosuits require external connection to a large and heavy air source, which makes them basically unportable.

**Twisted string actuator:** The basic architecture is formed with one end of two or more parallel strands fixed on the motor output shaft and the other end connected to the anchor point placed on the limb. Rotating the motor makes these strands intertwine with each other, consequently shortening their length and driving the joint to move. This mechanism enables the successful conversion from motor rotation to tendon contraction [116].

Twisted string actuators appear to be well-known for their simple and compact structure. Small-sized motors with high revolving speed and small torque are adequate for the power output tasks. The strands can completely replace the relatively complex reducer to perform the same function. These all benefit from the inherent property of a high reduction ratio. In addition, it can achieve a fairly high transmission efficiency [117].

Even so, the disadvantages of twisted string actuators cannot be ignored. Since the transmission ratio changes as the strands continue to twist, it adds to the difficulty of precise control. Repeated twisting and untwisting will aggravate fatigue damage, thus reducing its actual service life.

**Shape memory alloy-based actuators:** When exposed to a certain stimulus, the crystal arrangement tends to go through the phase transformation, causing the shape to change temporarily. Once this external intervention disappears, it gradually returns to the original memory state [118]. By actively applying or withdrawing stimuli, this actuator can be controlled to produce joint assistance as desired. Nickel titanium (NiTi) alloy belongs to the most commonly used material, which can be actuated by Joule heating.

Compared to other forms, shape memory alloy-based actuators are characterized by high power density and high tensile stress. However, although the contraction process is conducted quickly, it takes a lot of time to restore the original shape, resulting in a very low driving frequency. Some studies have attempted to utilize active cooling systems to improve actuation speed [43], but it increases the structural complexity for wearable robots. In addition, issues such as hysteresis and creep also adversely affect its performance.

**Passive actuator:** Passive elements mainly consist of springs and elastic fabrics. Arranged in parallel with muscles involved, they can store energy during one fixed phase of the limb’s movement and release it to generate assistance during another fixed phase [63].

Obviously, this approach makes the system extremely simple and lightweight. However, its effect on power assistance seems very limited considering that no additional energy can be generated.

**Quasi-passive actuation:** This refers to the combination of passive elements (namely springs, elastic cords or dampers), and quasi-passive elements (namely variable dampers or electromagnetic clutches). Exosuits using this method do not provide active torque at joints but can decide when passive assistance is deactivated or activated [119].

This configuration will make the system compact and lightweight, which is suitable for assisting multiple DoFs at the same time. Furthermore, it can utilize quasi-passive elements to adjust the period and magnitude of energy stored and released by passive elements in a planned way. The disadvantage is also that the effect of power assistance behaves in a relatively limited way.

### 3.3. Perception and Motion Control

Powered exosuits need to perceive the motion and state information of human–machine systems, based on which they can apply appropriate control algorithms to produce extra joint power. At first, we discuss related research on some typical control methods of upper extremity systems, which can be summarized as follows.

**Mirroring algorithm (mimetic control):** According to the reference input of a normal arm, this method can guarantee to help the target arm move to basically the same position. Auxilio adopted a Kinect Xbox 360 motion sensor device to measure the reference joint angle and the target one for making comparisons [48], while CRUX achieved a similar function via an IMU network [120].

**Gravity compensation control:** This algorithm will exert additional torques on human joints, compensating for the burden of limb gravity. The assistive values can be estimated according to the tension measured by force sensors. In [19,26], it was combined with velocity closed loop or an admittance controller to construct a more perfect motion control architecture.

**Adaptive control:** This strategy aims at automatically adjusting certain control parameters or laws so that it can adapt to dynamic changes in man–machine interaction. In [121], a model reference adaptive control made the performance of closed-loop controller basically consistent with that of a reference model through an adaption mechanism, to meet the tracking requirements for specific practice profiles of stroke patients. An adaptive assistive controller proposed in [122] tried to provide feedforward assistance with the inverse dynamic model learnt online.

**Backlash compensation control:** This method is applicable to cable-driven exosuits for improving the performance of motion control. Considering that the friction between cable and sheath will cause transmission delay, some researchers established a backlash hysteresis model and proposed two control strategies for compensation including a backlash observer-based controller and nonlinear adaptive controller to solve this problem [17]. In another article, a hierarchical control paradigm aimed at decoding the wearer’s motion intention at a high level, compensating backlash at the middle level, and driving the actuator at a low level [20].

**Biosignal-based control:** A promising method to control exosuits based on parsing human intentions from physiological signals. A neural network-enhanced torque estimation control strategy [123] was developed via combined application of surface electromyogram sensors, IMUs, load cells and encoders, which contained three functional parts including the neural-network adjustment module, the joint torque estimation module, and the closed-loop PID-based control module. Another sEMG-based controller relied on a double threshold strategy with a standard PID regulator to compensate the wearers’ muscle activities during a load carrying task [33]. In [35], a myoprocessor was designed based on an EMG-driven musculoskeletal model to estimate joint torque, which could guarantee adaptive assistance to the elbow movement when holding different loads. In subsequent research, the authors further analyzed how assistance magnitude will affect muscular benefits with the myoprocessor [124] and compared it with a force control [125], which proved the effectiveness and advancement of this method. Furthermore, a torque estimation-based control strategy [22] took advantage of mechanomyography to obtain expected output of joints, and then established a torque closed loop to achieve motion assistance.

Because of the obvious regularity of walking or running, the control of lower extremity systems reflects different design concepts. Such exosuits often detect gait events through IMUs or plantar pressure sensors/foot switches and customize force or position profiles to assist joints at the appropriate period of a gait cycle. Some key assistive parameters can be adjusted through optimization algorithms to enhance human–machine adaptability. For example, the strategy proposed in [126] took scaled reference force trajectories as input and implemented force following process with the help of an admittance controller. A Bayesian optimization attempted to adjust the peaking and offset time of force profile to minimize the metabolic cost for walking [127]. Of course, there were also control methods that did not rely on gait detection. Myosuit adopted an anti-gravity control strategy, which estimated the required assistive force based on the current posture, and provided gravity compensation effects to the joints with the force closed loop [93].

### 3.4. Evaluation Methods

After the wearable robot is integrated and debugged, it needs to undergo specific tests to demonstrate the actual performance and provide a reference for subsequent optimization. The focus of these assessments may change according to different application requirements. We intend to choose the common and general ones for discussion.

**Metabolic consumption:** This evaluation primarily illustrates the effect of using exosuits on the variation of the wearer’s own energy expenditure [60]. The specific value is obtained by indirect measurement with a portable gas analysis system. Much research has measured carbon dioxide and oxygen rate to calculate metabolic power through a modified Brockway equation [50,64]. A large number of experimental results have proved that exosuits show an ability to reduce metabolic cost to different degrees.

**Muscle activity:** Joint output is accompanied by activation of corresponding muscle groups, during which bioelectrical signals come into being and can be captured by surface electromyography sensors [128]. After filtering the original sequence, its eigenvalues (such as the root mean square [65]) will be extracted to characterize the degree of muscle activity. In general, wearing an exosuit could contribute to reducing activation and relieving fatigue for human muscles.

**Assistive torque/force:** These indicators directly reflect the output capability of exosuits acting on human joints, which can be measured and calculated by force sensors. A larger value means a more significant effect on joint strengthening [57].

**Kinematic analysis:** By recording the variation of joint angles, this assessment attempts to compare the trend and range of limb movement before and after the usage of an exosuit. Generally, IMU or 3D motion capture can be used to complete the relevant measurements [96]. These experiments indicate that the exosuit brings very little changes to the natural movements of wearers.

**Walking performance:** Apparently this test only applies to lower extremity systems. Through measuring and analyzing walking speed, walking distance, ground clearance and other parameters, the actual effect of gait recovery and ambulation training will be illustrated [129]. The exosuit has a significant positive effect on improving walking performance.

**Subjective evaluation:** This aims to analyze the user opinions after hands-on experience, such as ease of putting on and taking off, and wearing comfort. These relevant data can be collected through a combination of tests and questionnaires [102]. By counting the scores of each subjective indicator, the specific application performance of an exosuit is finally obtained.

## 4. Applications and Challenges

Soft wearable robots are designed to play an important role in a variety of application scenarios. The main aspects can be summarized as follows.

**Medical application:** In the field of health service, the large population of elderly and disabled has brought about a growing demand for medical rehabilitation. Lightweight and flexible exosuits can replace traditional methods to provide convenient daily assistance and regular training for these people [8].

**Military activity:** Application to the military field can significantly improve the maneuvering speed, load capacity and endurance limit of soldiers. This wearable device may become a mounting platform for more electronic equipment and individual weapons, consequently enhancing the abilities of combat and survivability [130].

**Manned spaceflight:** The field of manned spaceflight is also a potential target [131]. Exosuits can be adaptively integrated with spacesuits to help astronauts carry more scientific instruments, while delaying muscle fatigue and extending extravehicular activity time [132]. These basic functions will effectively improve the efficiency of outer space exploration.

**Industrial production:** Workers can perform manual labor such as lifting, carrying and assembling more efficiently with the help of exosuits [133]. They possess the ability to relieve physical fatigue and improve working efficiency [47]. Besides, some illnesses and injuries associated with overstrain could be prevented if proper assistance is provided.

**Personal usage:** In the future, exosuits may become a smart wearable device widely used in the daily life of normal people, which will provide power assistance for various aspects such as recreational sports and housework. It will be highly integrated with human intelligence to provide users with a high-quality life at the technical level.

However, the current research on soft wearable robots basically is in the laboratory stage. There are still many problems that prevent them from being widely applied. Some of the challenges are listed below.

The inherent characteristic of flexible power transmission increases the nonlinearity of the system and makes it more susceptible to disturbances. Therefore, it presents a challenge for precise motion control.In order to achieve multi-joint assistance in multiple DoFs, numerous actuators and complex structural layouts will be required. This is not conducive to a lightweight and compact design.Due to the special human-in-the-loop form and the obvious differences between most prototypes, the results of current performance test methods seem one-sided and imperfect. We need to build a unified standard evaluation framework to comprehensively reflect the actual effect and application value of every soft wearable robot.It is difficult to possess all excellent performance features such as portability, comfort, high efficiency and be lightweight at the same time. New or improved forms for power assistance need to be further developed.

## 5. Conclusions

In this paper, we have classified and discussed soft wearable robots for the upper limb, lower limb and back according to different joints, summarized the current usage of key technologies from the aspects of structure, actuator, control, and evaluation, and finally described the future applications and some problems hindering further development. This article tries to provide some references for future research in this field.

In conclusion, a series of soft wearable robots that assist different human joints have been designed for specific functional requirements in recent years. Relevant researchers have tried various methods to improve human–machine compatibility and assistive efficiency, which accumulate rich technical achievements in this process. The resulting prototypes may make outstanding contributions to many aspects of human life. However, the studies on soft wearable robots are still in the laboratory stage, which is far from practical application. We believe that future research may develop in several directions. Compatible and compact textile architectures should be further explored to make them similar to ordinary clothes. The development methods of lightweight and powerful actuators will become one of the research priorities, thereby raising the assistive efficiency of exosuits to a higher level. More comprehensive and efficient means of perception need to be proposed to achieve a perfect understanding of human intentions. More precise and compliant control strategies will be applied to enhance the movement consistency between human and machine.

## Figures and Tables

**Figure 2 sensors-22-07584-f002:**
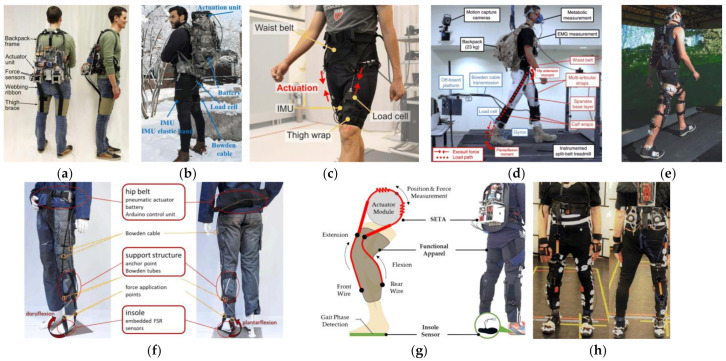
Exosuits for lower limbs: (**a**–**d**) Systems proposed by Wyss Institute for Biologically Inspired Engineering of Harvard University [49,50,51,52]; (**e**) Myosuit presented in [53]; (**f**) A cable-driven wearable device to support the ankle [54]; (**g**) Device with a series of elastic tendon actuators (SETA) to support stair ascent and descent [55]; (**h**) The Beta 1 prototype of XoSoft [56].

**Figure 3 sensors-22-07584-f003:**
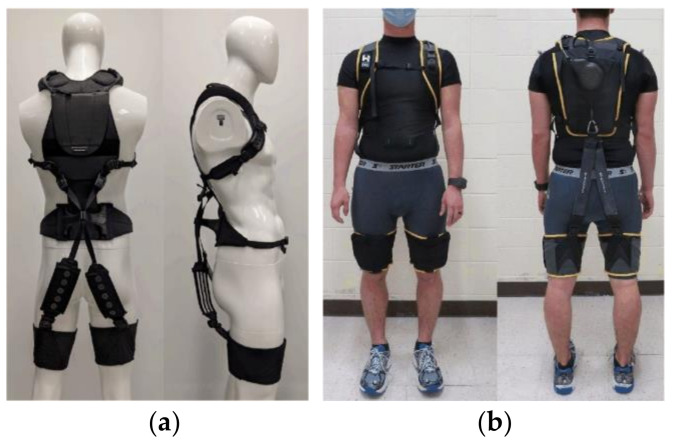
Back-assist exosuits: (**a**) A low-profile and dual-mode prototype using extension mechanism [104]; (**b**) HeroWear Apex [105].

**Table 6 sensors-22-07584-t006:** Several representative exosuits for knee assistance.

Exosuit/Study	Year	Movements	Actuation	Sensors	Control	Function
Hitexosuit [70]	2019	Extension	Twisted string actuator	IMU	Control based on gait period detection	Climbing stairs
Park et al. [71]	2020	Extension	Cable-driven module	IMU, load cell	Gait event detection, hyperextension protection, admittance controller	Healthcare
Lee et al. [72]	2020	Flexion/extension	Cable-driven module	IMU, insole sensor	Admittance controller	Climbing stairs
Sridar et al. [73]	2020	Extension	Pneumatic actuator	Pressure sensor, insole sensor	-	Physical enhancement
Fang et al. [74]	2020	Flexion/extension	Pneumatic actuator	Pressure sensor	On/off control algorithm	Healthcare
SLAK [75]	2021	Extension	Pneumatic actuator	Pressure sensor	Closed-loop control	Assistance for sit-to-stand

**Table 8 sensors-22-07584-t008:** Lower extremity exosuits for multi-joint assistance.

Exosuit/Study	Year	Movements	Actuation	Sensors	Control	Function
Wehner et al. [91]	2013	Assistance for hip, knee and ankle	Pneumatic actuator	Footswitch, pressure gauge	Timing based control scheme	Physical enhancement
Quinlivan et al. [92]	2017	Hip flexion, ankle plantarflexion	Cable-driven module, passive element	Load cell, gyroscope	Biologically inspired control	Physical enhancement
Myosuit [93]	2017	Flexion/extension for hip and knee	Cable-driven module, passive actuator	Load cell, encoder, IMU	Posture based anti-gravity control	Physical enhancement
Lee et al. [1]	2018	Hip flexion and extension, ankle plantarflexion	Cable-driven module, passive element	Load cell, IMU	Force-based position control	Military application
Beta 1 prototype of XoSoft [56]	2019	Flexion of hip and knee	Quasi-passive actuation	Shoe insole	Control based on gait segmentation	Healthcare
Gamma prototype of XoSoft [94]	2020	Assistance for hip, knee and ankle	Pneumatic quasi-passive actuation	Insole sensor, IMUs, pressure sensor	Gait cycle segmentation, finite state machine	Healthcare

## Data Availability

Not applicable.

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
