# Peer review of "Soft Wearable Robots: Development Status and Technical Challenges"

_sensors, 2022, doi:10.3390/s22197584_

Round 1

Reviewer 1 Report

This review paper focused on power-assisted wearable robotic device. The review follows a logic that is to describe/categorize the research interest in the society by different human joint/tasks to assist, followed by key technologies/challenges in recent studies. Overall I think the review can be easily followed by general/a broad audience. There are however some points that can be revised to further improve the quality of this paper.

1. What would be a commonly agreed definition of "power-assisted"? What does the "power" mean and what is the conceptual scope of "assist"? Speaking of the "power", for example, should (hybrid) neuroprosthesis that uses external electrical stimulation to recruit muscle's energy for joint actuation thus in some situation (assisting/recovering impaired motor function in patients or for normal healthy person, amplifying muscle's power recruitment compared to the volitional intend/effort) be included in the scope of the power-assisted robotic device (human-in-the-loop)? Speaking of the "assist", does the robotic devices for amputee include in the scope of the review. A more clear clarification from the beginning paragraphs can help to better define the scope of the review.

2. It is suggested that more interesting, recent advanced technical details can be added in the review. For example, in general, in addition to the descriptive introduction on the structure/mechanical designs, what are the different robotics models/human-in-the-loop (joint, limb) models (mathematically) used for those mentioned different wearable devices?

3. The same comment as 2. applies when talking about the control strategy, especially section 3.3. The current contents are too general and vague. It is suggested that, the authors can present the technical details of those control algorithms used in the selected refences and then provide their own comments (pros and cons in terms of stability, robustness, response time, power consumption, etc.).

I think the summary of these can greatly benefit people in robotics/control society and improve the educational contribution of this review paper. 

Reviewer 2 Report

General comments:

This manuscript aims to review the development status and technical challenges of powered soft wearable robots. The review grouped the powered soft wearable robots by the body parts that support, such as upper and lower extremities exosuits. The review also summarized several key technology aspects—structure, actuator, control and evaluation. In general, the manuscript provided a good overview of the current state of powered soft wearable robots and covered several existing applications and challenges of this emerging technology. However, back-assist wearable robots are missing in this review, which are in high demand for industrial and rehabilitation applications. It would be nice to include the current states of back-assist exosuits. In addition, some sentence structures and word choices still need more work. Last but not least, the reviewer is not familiar with the copyright policy, but do the authors need to get copyright clearance for the cited figures from previous publication?

Title: Consider changing to “Powered Soft Wearable Robots…”

1.       Introduction.

In general, consider using some exoskeleton definitions and terms from ASTM F48 (ASTM F48 Formation and Standards for Industrial Exoskeletons and Exosuits - PubMed (nih.gov))

Line 65: consider changing to “we aim to comprehensively review the current …”

Line 68-71: Back-assist soft wearable robots were completely left out in this review paper, although it said all soft wearable systems were divided into upper and lower extremity exosuits.

Line 76-78: consider rephrasing this sentence (maybe combine them into one statement)?

2.       Overview of Exosuit

Line 83-84: consider changing to “help a user walk or climb stairs”

Line 84-85: consider changing to “There are differences in the anatomical structures and movement patterns for each joint.”

Line 85-86: consider changing to “Different assistance schemes should be carefully evaluated according to specific application requirements.”

Line 111-116: As to the order of the pictures in Figure 1, is it possible to follow the order in the paragraph above (Line 97 to 106)—the picture of ref [11] comes first and then ref [12] and [13]? Also, in Figure 1 (b), what are “ABA” and “HEFA”? If they are not discussed in the manuscript, they should be removed.

Line 117: consider changing to “assistance for multiplanar movements of shoulder…”

Line 154: spell out “PWM” for the first time

Line 160: consider changing to “a powered elbow exosuit”?

Line 167: consider changing to “built a soft pneumatic elbow exosuit”

Line 170: spell out “EMG” for the first time

Line 176: spell out “sEMG” for the first time

Line 221: consider changing a different subtitle. Maybe “multi-joints in upper extremity”?

Line 223: consider choosing a different word for “momentum”

Line 275: consider changing to “achieving significant reduction…”

Line 299: consider adding the lab name of Harvard University in Figure 5 caption

Line 305: the statement is not accurate, consider changing to “As the largest joint and one of the most complex ones in human body, the knee…”

Line 336: consider changing to “Several representative exosuits for knee joint…”

Line 365: consider changing to “Several representative ankle exosuits …”

Line 368-369: Does the exosuit in Figure 7f fit the topic since we are talking about powered exosuit here?

Line 384: similarly, consider changing to a different subtitle. Maybe “multi-joints in lower extremity”?

3.       Key Technologies

Line 438: consider changing to “Material and Structure”

Line 452: consider changing to “Efforts should be made to guarantee …”

Line 474: suggest citing references for the “cable-drive module”

Line 525: consider using “passive actuator” not “passive element” in the beginning of the paragraph

Line 598: consider changing to “3.4 Evaluation Methods”

Line 615-627: suggest citing references for these paragraphs since this is a review paper…

Reviewer 3 Report

The research topic is interesting. However, the current draft is not acceptable for publication. Both Abstracts and Conclusions are easy for readers to understand this work but they were not fully supported by provided discussions in the main manuscript. The writing quality is good but the organization of this review must be improved. This study overviews the development status of soft wearable robots for human movement assistance. The authors perform a system classification according to the target-assisted joint and attempt to describe the overall prototype design level in related fields. They discuss the possible application fields and extract the main challenges of this valuable research direction. A few suggestions which can further improve the quality of this work are

1.     Please recheck all abbreviations and define them in the first place of appearance as a few abbreviations are not described such as PID, EMG, PWM, etc.

2.     Please provide the organization of this study.

3.     It will be interesting to add a comparative Table in the Introduction section and compare your work with existing reviews or surveys.

4.     It is suggested to provide a precise summary at the start of each section (as you provided in Section 3) to define contributions in it.

5.     Even though figures are properly labeled. However, the quality of the text inside is bad. It does not make sense to crop and put figures. Try to improve the presentation and resolution quality of figures.

6.     Why you did not mention figures within relevant sections? The figures must be cited properly within the Text.

7.     I suggest authors must add a comprehensive Table in Subsections 3.3 for different sensing and motion control approaches.

8.     There is a complete lack of discussion and references in Subsection 3.4.

9.     What’s the point of adding Applications and Challenges without any reference and proper discussion? The authors should provide a comprehensive discussion along with references.

10.                        It will be interesting to provide an attractive figure for its applications.

11.                        A review without incorporating Tables is incomplete. From the Tables, the reader can fastly understand the research contributions. Try to add some comparative parameters such as components, tools, advantages, tasks, application domain, etc.

12.                        The Conclusion must be revised. Also, add a few possible future research directions in this domain.

13. Moreover, what was your criteria or approach to select relevant studies?

Round 2

Reviewer 1 Report

No further comment

Reviewer 2 Report

The authors have addressed the reviewer's comments.

Reviewer 3 Report

Dear authors,

Thank you for your great efforts. You have addressed my concerns adequately. This revised draft is much better in quality.

All the best